# Linezolid Serum Concentration Variability Among Critically Ill Patients Based on Renal Function and Continuous Renal Replacement Therapy Administration

**DOI:** 10.3390/antibiotics14121188

**Published:** 2025-11-21

**Authors:** Stefano Agliardi, Beatrice Brunoni, Gianluca Gazzaniga, Leonardo Baggio, Riccardo Giossi, Greta Guarnieri, Stefania Paccagnini, Matteo Laratta, Thomas Langer, Sara Santambrogio, Gianpaola Monti, Romano Danesi, Francesco Scaglione, Arianna Pani, Roberto Fumagalli

**Affiliations:** 1Department of Medical Biotechnologies and Translational Medicine, Post-Graduate School of Pharmacology and Clinical Toxicology, University of Milan, 20122 Milan, Italy; gianluca.gazzaniga@uniroma1.it (G.G.); greta.guarnieri@unimi.it (G.G.); 2Chemical-Clinical Analyses, Poison Control Center and Clinical Pharmacology Unit, ASST Grande Ospedale Metropolitano Niguarda, 20162 Milan, Italy; riccardo.giossi@ospedaleniguarda.it (R.G.); stefania.paccagnini@ospedaleniguarda.it (S.P.); romano.danesi@unimi.it (R.D.); 3Department of Medicine and Surgery, University of Milano-Bicocca, 20126 Milan, Italy; b.brunoni@campus.unimib.it (B.B.); l.baggio6@campus.unimib.it (L.B.); thomas.langer@ospedaleniguarda.it (T.L.); roberto.fumagalli@unimib.it (R.F.); 4Department of General Surgery, Surgical Specialty and Anesthesiology Paride Stefanini, Sapienza University of Rome, 00161 Rome, Italy; 5Department of Anesthesia and Intensive Care Medicine, ASST Grande Ospedale Metropolitano Niguarda, 20162 Milan, Italy; matteo.laratta@ospedaleniguarda.it (M.L.); saramicol.santambrogio@ospedaleniguarda.it (S.S.); gianpaola.monti@ospedaleniguarda.it (G.M.); 6Department of Oncology and Hemato-Oncology, Università degli Studi di Milano, 20122 Milan, Italy; francesco.scaglione@unimi.it (F.S.); arianna.pani@unimi.it (A.P.)

**Keywords:** linezolid, antimicrobials, therapeutic drug monitoring, intensive care, pharmacokinetics

## Abstract

Background: Linezolid standard dosing is typically applied in ICU without adjustments, even in renal impairment. This study examines serum concentration variability by renal function or CRRT administration in patients receiving 1200 mg/day of linezolid. Methods: This retrospective, single-center, non-randomized observational study was conducted at Niguarda Hospital (Milan, Italy) on data from the two-year period 2023–2024. ICU patients receiving linezolid, with a renal function determination and trough TDM performed at steady-state were included and stratified by renal function or CRRT status. Results: 54 patients were included, with 18 (33%) undergoing CRRT (CVVH). CRRT patients presented higher median linezolid concentrations (4.6 mg/L) than non-CRRT patients (3.2 mg/L), and a lower risk of underdosing (17% vs. 39%). CRRT patients showed significantly lower concentrations (4.6 mg/L vs. 10 mg/L, *p* = 0.007) than non-CRRT patients with renal function ≤ 30 mL/min, with fewer out-of-range levels (39% vs. 91%, *p* = 0.008) and overdosing (22% vs. 73%, *p* = 0.018). A significant inverse correlation was found between renal function and linezolid levels (Spearman’s rho = −0.61, *p* < 0.001), with CRRT patients exhibiting concentrations comparable to those of individuals with moderately impaired renal function. Continuous infusion resulted in significantly higher median concentrations (7.2 mg/L) than extended infusion (2.7 mg/L), with an increased risk of overdosing (47% vs. 17%; *p* = 0.018). Conclusions: After standard-dosing administration, linezolid levels vary widely in critically ill patients. Renal function significantly affects pharmacokinetics: severe impairment increases overdose risk, while ARC may cause underdosing. Standard-dosing appears adequate in CRRT patients, with levels similar to moderate-impairment. Continuous infusion aids target attainment in normal or ARC patients but raises overdose risk in severe impairment. TDM-based personalized dosing seems crucial to optimize therapy and reduce toxicity or failure.

## 1. Introduction

Linezolid is an oxazolidinone antibiotic frequently used in intensive care units (ICUs) to manage Gram-positive bacterial infections. The standard adult dose is 600 mg twice daily, and no dosage adjustments are foreseen even in the case of patients with significant alterations in renal or hepatic function [1]. Anyway, multiple factors commonly observed in critically ill patients can represent a challenge in achieving optimal pharmacokinetics/pharmacodynamics (PK/PD) targets, including increased vascular permeability, fluid shift, and organ dysfunction, which can lead to substantial variability in serum drug concentrations. This variability increases the risk of toxicity, therapeutic failure, and the emergence of antibiotic resistance [2].

Renal function often retains a crucial impact on the pharmacokinetics of antibiotics, especially for those predominantly eliminated by the kidney [2]. Linezolid itself also undergoes renal elimination, with 30–40% of the administered dose excreted via the kidneys, of which approximately 30% is excreted as the unchanged drug [1]. Chronic renal failure is common, especially among elderly and frail individuals, and is associated with an increased risk of admission to ICUs [3]. In critically ill patients, renal function may decline, particularly in cases of sepsis, ranging from mild to moderate impairment to severe renal insufficiency and acute kidney injury (AKI) [4]. Continuous renal replacement therapy (CRRT) represents a cornerstone treatment for severe acute kidney injury (AKI). However, CRRT filtration capacity varies depending on several factors, including the type of machine, filter characteristics, blood and dialysate flow rates, and infusion modality. Similar multifactorial variability also exists for CRRT impact on the clearance of several drugs, including linezolid [5]. Moreover, some ICU patients may also present augmented renal clearance (ARC), which accelerates renal drug elimination, complicating therapeutic management [6].

Therapeutic drug monitoring (TDM) has been proposed to ensure adequate serum concentrations and prevent both overdosing and underdosing [7]. The hypothesis is that the standard dose of linezolid may not be suitable for all patients, regardless of their clinical and demographic conditions. To the best of our knowledge, no previous study has comprehensively described TDM of linezolid in critically ill patients, with and without CRRT. The aim of this study is to describe the variability of linezolid serum concentrations in critically ill patients, in relation to renal function and/or CRRT administration.

## 2. Results

During the study period, 937 patients were admitted to ICU. Of these, 193 (21%) received linezolid, and the TDM was performed in 64 cases (33%). Among the latter, 54 patients (84%) met the inclusion criteria, with 18 of them who underwent CRRT (33%). CRRT and non-CRRT populations were comparable in terms of key variables such as sex, age, body mass index (BMI), albumin levels, and liver function. In 20 cases (37%), linezolid was prescribed as targeted therapy against specific pathogens (*Enterococcus* spp. or *Staphylococcus* spp.). None of these isolates had a linezolid MIC greater than 2 mg/L, and following an initial positive culture, all patients converted to negative and were successfully discharged from the ICU in due course.

The general characteristics of the study population are summarized in Table 1.

Among the included population (N = 54), 32 patients (58%) presented out-of-range C_min-ss_ linezolid serum concentrations. Specifically, 15 patients (27%) experienced overdosing, while 17 patients (31%) experienced underdosing.

Overall, patients undergoing CRRT exhibited higher, although non-significant, median linezolid serum concentrations compared to non-CRRT patients, with greater adherence to therapeutic ranges (61% vs. 31%, *p* = 0.031) [Table 2A]. Patients undergoing CRRT showed significantly lower median linezolid serum concentrations compared to non-CRRT patients with renal function ≤ 30 mL/min. They also presented a lower frequency of out-of-range concentrations (39% vs. 91%, *p* = 0.008) and overdosing occurrences (22% vs. 73%, *p* = 0.018) [Table 2B].

A significant inverse correlation was observed between renal function classes and plasma linezolid concentrations (Spearman’s rho = −0.61, *p* < 0.001), indicating that decreasing renal function was moderately associated with higher linezolid levels [Figure 1A]. Linezolid serum concentrations in patients undergoing CRRT appear to fall between those observed in the ≤30 mL/min group and the 31–70 mL/min group. Patients in the ≤30 mL/min group are exposed to the highest risk of overdosing (72.7%) [Figure 1B and Appendix A].

Patients who received linezolid via continuous infusion showed significantly higher serum concentrations compared to those who received it via extended infusion. The risk of overdosing was significantly higher (*p* = 0.018) in patients receiving linezolid via continuous infusion compared to those with extended infusion (47% vs. 17%) [Table 3A].

Compared to extended infusion, higher linezolid serum concentrations determined by continuous infusion were evident even when CRRT patients and non-CRRT patients are analyzed separately, with a statistically significant difference among the latter (*p* = 0.003) [Table 3B and Appendix A].

## 3. Discussion

The present study analyzed serum linezolid concentrations in critically ill patients, performing comparisons based on renal function and the use of CRRT. This work specifically highlights an inverse correlation between renal function and serum linezolid levels, a higher risk of overdosing in patients with end-stage renal disease compared to those receiving CRRT or with preserved renal function, and increased serum concentrations when the drug was administered via continuous infusion.

Due to its pharmacokinetic characteristics, various clinical conditions may influence the serum concentrations of linezolid, particularly among critically ill patients. One of the most relevant factors was renal function. ICUs often treat patients with a broad range of renal functions, from mild to moderate renal insufficiency to severe impairments in glomerular filtration, not to mention individuals with ARC, while some patients also requiring CRRT. In these cases, estimating the impact of such treatment on the pharmacokinetics of many drugs, including linezolid, becomes particularly challenging.

Although dose adjustments for linezolid are not recommended in the case of renal impairment, a known correlation exists between glomerular filtration rate, linezolid clearance, and its serum concentrations. Sasaki et al., with their population model, demonstrated that renal failure and severe liver insufficiency can significantly alter the pharmacokinetics of linezolid. Using a mathematical model based on collected data, the authors simulated various scenarios and proposed reducing the dose in patients with reduced renal filtration (particularly those ≤30 mL/min) or severe hepatic insufficiency (Child-Pugh C). In these patients, drug concentrations can easily reach toxic levels, directly correlating with the onset of severe side effects such as thrombocytopenia. Thus, this study emphasizes the importance of personalizing the linezolid dose based on the patient’s clinical condition, to optimize therapy and enhance safety and effectiveness in patients with severe renal or hepatic dysfunctions [8]. Our study highlights the potential risk of deviating from the therapeutic window in critically ill patients undergoing linezolid therapy with a standard dose of 600 mg twice daily, both in terms of overdosing and underdosing. This risk appears closely related to the patient’s renal function, with a higher risk of overdosing in patients with severely impaired renal function (and especially with a GFR ≤ 30 mL/min) and a higher risk of underdosing in subjects with ARC (GFR > 120 mL/min), suggesting a proportional inverse relationship between the patient’s renal function and linezolid serum concentrations.

Among acutely ill patients, eGFR may substantially overestimate true renal function due to reduced creatinine production and load [9], delayed creatinine accumulation in the setting of rapidly changing renal function [10], and hemodilution associated with fluid resuscitation or volume overload [11]. Because accurate assessment of renal function is essential for optimizing linezolid dosing, therapeutic decisions should rely preferentially on directly measured creatinine clearance rather than eGFR alone. Indeed, formulas used to estimate renal function often perform poorly in critically ill populations and should therefore be applied cautiously in clinical decision-making [12,13]. In patients with impaired or unstable renal function, dose individualization of linezolid is reasonable, and TDM may help ensure that plasma concentrations remain within the therapeutic range [14].

In the context of the CATCH study, Corona A. et al. assessed the Sieving coefficient of various antibiotics, including linezolid. Their findings revealed that CRRT actively removes a wide array of antibiotics, depending on the drug’s properties, machine settings, and membrane type. Specifically, antibiotics with a low volume of distribution (Vd < 1 L/kg), weak plasma protein binding, and predominantly renal clearance seem particularly prone to be removed. These results also suggest a significant impact of CRRT on linezolid pharmacokinetics and underscore the critical role of TDM in managing antimicrobial therapy among critically ill patients [15]. In a recent systematic review by Liu Y., et al., the impact of CRRT on the clearance and serum concentrations of linezolid was examined. The authors identified only six studies investigating the pharmacokinetics of linezolid in CRRT patients. These studies included different populations, varying in sample size (minimum 17, maximum 27), CRRT type, and control groups (one study compared CRRT patients with non-CRRT patients, while another compared CVVH with intermittent hemodialysis). In all of these studies, linezolid was administered at a standard dose of 600 mg twice daily by short infusion (30–60 min). The authors noted significant inter-individual variability in linezolid serum concentrations among CRRT patients, with CRRT significantly contributing to drug removal. These findings suggest that the conventional dosing regimen for CRRT patients could be insufficient for those with residual renal function but overdosed for anuric patients. Therefore, the optimization of linezolid dosing regimen based on TDM, which includes dose adjustments, different dosing intervals, and alternative infusion modalities, is essential in CRRT patients to achieve the therapeutic goal. This approach is particularly important in patients with high body weight, preserved renal function, and when the minimum inhibitory concentration (MIC) of the treated bacteria exceeds 2 mg/L [16]. Besides molecular and clinical variables, in their systematic review, Villa et al. investigated the effect of CRRT on linezolid pharmacokinetics, depending on the different modalities and types of membrane. In this context as well, the authors noted strong interindividual variability, suggesting the use of TDM strategies to personalize linezolid dose. Specifically, despite this high variability, the authors did not identify a significant difference in linezolid removal between different techniques (diffusive vs. convective), while emphasizing a strong impact of filter pore size and membrane composition [17]. Patients receiving CRRT in our study were comparable in terms of treatment modality and dosing regimen. Therefore, the CRRT population can be considered a distinct and homogeneous group. Nonetheless, it should be acknowledged that not all extracorporeal treatments are equivalent—especially those involving adsorption techniques, for which a therapeutic drug monitoring (TDM) strategy may be crucial. In our cohort, patients undergoing CVVH did not appear to be at a higher risk of overdosing or underdosing compared to the general population of critically ill patients. However, the risk was significantly lower compared to that of patients with end-stage renal failure (GFR ≤ 30 mL/min) who did not receive renal replacement therapy. Regarding renal function, critically ill patients undergoing CVVH seemed to have serum linezolid concentrations similar to those of subjects not undergoing CVVH but with partially preserved renal function (with a median between the groups ≤ 30 mL/min and 31–70 mL/min), presenting a similar risk profile for exposure to overdosing or underdosing. Although some interindividual variability exists and the risk is not negligible, in our cohort of CVVH patients, the standard dose of 600 mg twice daily seemed adequate in most cases in terms of target attainment (2–7 mg/L). Conversely, this regimen was insufficient to consistently reach target plasma concentrations in subjects with a GFR > 70 mL/min, particularly among those with a GFR > 120 mL/min.

In the context of potential linezolid dosing adjustments, other major clinical variables such as liver function and albuminemia may have a significant impact, in addition to renal function. Liver dysfunction is observed in up to 20% of patients admitted to the ICU. When abnormalities in liver tests are detected, a systematic and targeted evaluation should be undertaken to distinguish between primary hepatic injury (e.g., acute liver failure or acute-on-chronic liver disease) and secondary hepatic dysfunction resulting from critical illness or therapeutic interventions. The interpretation of liver biochemistry tests (LBTs)—typically including bilirubin, AST, ALT, alkaline phosphatase, gamma-glutamyl transferase, serum albumin, urea, and coagulation indices—should be contextualized within the overall clinical picture. This distinction is essential not only to guide appropriate supportive management but also to assess the liver’s capacity for compensation and recovery. In patients without pre-existing liver disease, secondary hepatic dysfunction is the most common cause of abnormal LBTs in the ICU, often related to systemic inflammation, hemodynamic instability, sepsis, or drug exposure. In such cases, resolution of the underlying cause frequently results in normalization of hepatic parameters [18]. In our study, although mild elevations in liver function markers were observed, no patient developed clinically significant hepatic injury or liver failure. To date, even when liver function alterations occur, the SmPC does not recommend dosage adjustments [1], although some evidence in the literature suggests that the impact may be clinically relevant, at least in patients with Child-Pugh class C disease [19]. Hypoalbuminemia was also frequent across all groups, consistent with what is typically observed in critically ill patients. This condition likely reflects a combination of systemic inflammation, increased capillary permeability, and/or reduced hepatic synthesis [20]. Although linezolid pharmacokinetics could theoretically be influenced by low albumin levels—potentially increasing the free drug fraction, drug clearance, and Vd—no studies have clearly demonstrated a clinically significant impact [21], which seems unlikely unless severe hypoalbuminemia (<2 g/L) occurs.

Although continuous infusion is not recommended, the stability of linezolid has been demonstrated for more than 24 h in saline NaCl 0.9% or glucose 5–10% solutions [22]. The recent consensus on the administration of time-dependent and AUC-dependent antimicrobials such as linezolid, particularly in high-care settings such as ICUs, suggests the use of continuous and/or extended infusions to achieve optimal PK/PD target attainment [23,24]. Warda et al., in the currently largest comparative study (179 patients) between the two methods of administration, demonstrated the superiority of continuous infusion over extended infusion in terms of efficacy (clinical cure rate, improvement in P/F ratio, incidence of sepsis after treatment initiation, and time to clinical cure) as well as safety (lower incidence of thrombocytopenia) [25]. In a more recent study, Hui et al. further investigated these findings with pharmacokinetic analysis, observing greater variability in linezolid serum concentrations in patients receiving intermittent infusions compared to those on continuous infusion. Continuous infusion showed better performance in achieving targets such as T > MIC > 80% for MICs of 1 and 2 mg/L and AUC24–48/MIC > 80 for an MIC of 1 mg/L. Based on these data, the authors emphasize that continuous infusion may offer potential benefits, including avoiding periods of suboptimal concentrations, which can improve safety profiles and clinical outcomes, providing a better opportunity to achieve PK/PD indices, even for MICs greater than 1 mg/L [26]. On the other hand, some authors have suggested that, to ensure an adequate AUC target even for MIC ≥ 2 mg/L, higher reference ranges should be followed when linezolid is administered by continuous infusion [27]. Pea et al. suggested optimal PK/PD targets of a T > MIC ≥ 85% of the dosing interval or an AUC/MIC > 80, and defined adequate serum concentrations between 2 and 7 mg/L to achieve these targets when linezolid is administered by intermittent infusion [28]. Although continuous infusion appears to be superior in achieving the first objective, it is possible that the target AUC may not be reached at standard doses with this administration mode, referring to the conventional reference range. However, current evidence confirms an increased risk of toxicity for linezolid serum concentrations exceeding 7 mg/L, regardless of the infusion method [27,29], whereas so far no data have demonstrated reduced effectiveness in terms of clinical and microbiological outcomes, once steady-state concentrations remain above the MIC. The most recommendable approach at the moment, therefore, seems to be the TDM-based one, with dosage adjustments and individualized targets based on various factors, such as high MICs or difficult-to-reach infection sites [27,28]. In our cohort, 75% of subjects treated via continuous infusion achieved linezolid serum concentrations ≥ 4 mg/L at steady state, both among patients undergoing CVVH and those not, demonstrating that this infusion modality may be promising in ensuring 100% T > MIC and providing higher drug levels than intermittent infusion. This approach can be advantageous for most subjects, but particularly for critically ill patients with preserved renal function (71–120 mL/min) or even ARC (>120 mL/min). On the other hand, this method may increase the risk of overdosing in patients with terminally compromised renal function (≤30 mL/min).

The limitations of this study include its retrospective and monocentric design. The sample size is relatively small; however, no studies currently available in the literature on linezolid and CRRT surpass the sample size analyzed here. The presence of only CVVH patients in this study is a limitation, but it ensures a clear criterion of comparability among the CRRT subjects analyzed, minimizing the risk of confounding. Subjects included in this work were also comparable for multiple other critical variables, such as liver function, sex, weight, and albumin levels. A strength of the study is the precise definition of renal function levels, as all of them were assessed using a reliable and accurate method, such as direct measurement of creatinine clearance.

Despite accumulating evidence highlighting the impact of renal dysfunction on linezolid exposure, the Summary of Product Characteristics (SmPC) and several current guidelines still assert that dose adjustments are unnecessary, even in the presence of substantial alterations in renal function. Based on the available data, initiating therapy with the standard regimen and subsequently individualizing dosing through early TDM, whenever feasible, seems reasonable. While additional high-quality evidence is still needed to guide evidence-based decisions and ensure truly individualized therapy from treatment initiation, in settings where TDM is unavailable, referral to centers with dedicated clinical pharmacology expertise could be considered; otherwise, intensified clinical and laboratory surveillance should be performed to detect early signs of toxicity, such as thrombocytopenia. Finally, while both intermittent and continuous infusion strategies appear clinically acceptable, continuous infusion may offer advantages in patients with augmented renal clearance and should be applied cautiously in those with impaired renal function—always within a TDM-driven framework.

## 4. Materials and Methods

### 4.1. Study Design

This is a retrospective, single-center, non-randomized observational study conducted at the two ICUs of Niguarda Hospital (Milan, Italy). The study includes patients receiving linezolid therapy for whom serum drug monitoring was performed at steady-state.

### 4.2. Objectives

Primary objective was to evaluate the variability of linezolid serum concentrations based on renal function and CRRT administration. The primary endpoint was the difference in serum concentrations between patients on CRRT and those not on CRRT. Secondary objectives included evaluating differences in serum levels based on administration methods (continuous infusion vs. extended infusion).

For each patient, a trough serum concentration of linezolid was determined upon reaching steady state (estimated after at least 36 h of therapy). Renal function was determined by direct measurement of 24 h creatinine clearance (CrCl), using the following formula:CrCl (mL/min) = (Ucr × UV)/SCr
where UCr stands for urinary creatinine (mg/dL), UV for urine volume (mL/min), SCr for serum creatinine (mg/dL).

Patients were subsequently classified based on renal function groups (≤30 mL/min, 31–70 mL/min, 71–120 mL/min, >120 mL/min) or CRRT.

Target attainment for serum linezolid concentrations was defined as 2–7 mg/L, both for intermittent and continuous infusion regimens, while a linezolid serum concentration > 7 mg/L has been associated with increased risk of potentiality severe adverse events such as high-grade thrombocytopenia, both in case of intermittent [29] and continuous infusion [27], without demonstrated benefits in terms of clinical efficacy or better clinical outcomes, thus this level should not routinely be exceed, regardless of the mode of administration [27]. Additionally, the MIC of linezolid for most staphylococci is 2 mg/L [30], which is commonly included as an objective lower limit that should not be undercut in various studies that investigated both intermittent and continuous infusion regimens [27,29,31,32]. Concentrations outside this range were defined as underdosing/overdosing.

### 4.3. Population and Inclusion Criteria

We included patients aged ≥ 18 years, admitted to the ICU from 1 January 2023 to 31 December 2024, who received linezolid upon clinical need and decision, with trough serum measurement performed upon reaching steady state (at least 36 h of therapy), and available valid renal function parameters (CrCl) at the time of TDM. Patients without inclusion criteria, unavailable data, or linezolid administration for less than 36 h were excluded.

A standard dose of 1200 mg/day of linezolid was administered intravenously to all subjects. In accordance with ward clinical practice and medical decisions, linezolid may have been administered either as an extended infusion (3 h) or as a continuous infusion (preceded by a 600 mg bolus administration). After at least 36 h of uninterrupted therapy with linezolid, adequate steady-state condition was considered to be reached. For patients undergoing multiple TDM measurements, only the first measurement for each patient was considered. In cases of multiple therapy cycles with linezolid, subjects were regarded as naïve patients only if more than 14 days had passed since the previous cycle, and new updated data were collected.

Demographic and clinical variables obtained from patients’ medical records, clinical diaries, and/or the laboratory portal were also collected for every subject. The main clinical covariates included BMI (kg/m^2^) and serum albumin (g/dL).

All CRRT patients were treated with continuous veno-venous hemofiltration (CVVH, 100%). CVVH was performed using the PrisMax equipped with AN69 ST150 hollow fiber filter (Baxter International Inc., Baxter Healthcare SA, Zurich, Switzerland). Diluted citrate (Regiocit, trisodium citrate 18/0, Baxter Healthcare Corporation, Deerfield, IL, USA) was administered for regional citrate anticoagulation (RCA). Among them, 11 (61%) maintained residual diuresis (mean = 966 mL/day), while 7 (39%) presented complete anuria. The machine settings and specific parameters are summarized in Appendix A. Q_uf_ (ultrafiltration rate) was defined as citrate flow + reinfusate flow − effluent. None of the non-CRRT patients received any type of renal function support.

### 4.4. Laboratory Measurements

Serum linezolid concentrations are routinely measured in our laboratory through high-performance liquid chromatography coupled with ultraviolet detector (HPLC-UV) using the Chromsystems reagents’ kit for antibiotics in serum/plasma for HPLC (Chromsystems Instruments & Chemicals GmbH, Gräfelfing, Germany), applied to the Shimadzu Nexera LC-40 with UV detector SPD 40V (Shimadzu Corporation, Kyoto, Japan).

Before injection into the chromatograph, biological samples (50 μL) were aliquoted into 1.5 mL vials, added with 10 μL of priming solution (hydrochloric acid) and mixed. The resulting supernatants were added with 100 μL of internal standard (Chromsystems 61004) and centrifuged again (15,000× *g* for 5 min). 50 μL of supernatant were then added with 50 μL of dilution buffer and mixed again. The preparations obtained were finally injected and separated on a reversed-phase C18 chromatographic column (Chromsystems 61100): the procedure was performed in isocratic elution, with an injection volume of 10 μL and a mobile phase (Chromsystems 61002) flow rate of 1.0 mL/min. The column temperature was maintained at +22 °C. The detector was set to a wavelength of 252 nm, appropriate for the specific optical absorption of linezolid.

### 4.5. Statistical Analyses

Descriptive statistics were used to summarize clinical and demographic data. Continuous variables were summarized using means and standard deviations or medians with interquartile ranges (IQR), according to distribution, while categorical variables were reported as absolute frequencies and percentages. Continuous variables were compared between groups using either the independent samples *t*-test or the Wilcoxon rank-sum test, depending on data distribution. For categorical variables, we used the chi-square test or Fisher’s exact test, as appropriate.

Data were first compared between CRRT and non-CRRT patients. An additional comparison was performed between patients with severe renal impairment undergoing CRRT and not undergoing CRRT (GFR < 30 mL/min). We also compared serum concentrations according to the type of administration (continuous infusion vs. extended 3-h infusion). Finally, Spearman’s rank correlation was used to assess the relationship between renal function categories and plasma linezolid concentrations. Data analysis was performed using R (version 4.2.3) software, with a two-sided significance threshold set at α = 0.05.

## 5. Conclusions

The standard daily dose of linezolid appears to be generally adequate for patients with partially preserved renal function and, similarly, for those undergoing CRRT, though there is some inter-individual variability, and the risk of deviating from the therapeutic range is not null. Serum linezolid concentrations are highly variable in critically ill patients after standard-dose administration, with a significant risk of falling outside therapeutic levels. An inverse proportional relationship exists between the renal function of critically ill patients and their serum linezolid levels, with a considerable risk of overdosing in patients with severely impaired renal function and a significant risk of underdosing in patients with ARC. Continuous infusion may be beneficial, especially for patients with higher renal clearance, but could increase overdose risk in patients with severe renal impairment.

Implementing precision medicine strategies through TDM, along with alternative infusion modes and dosing regimens, can be crucial to ensure adequate achievement of PK/PD targets. This approach would optimize therapy for critically ill patients, improve clinical outcomes, and minimize the risk of emerging bacterial resistance.

## Figures and Tables

**Figure 1 antibiotics-14-01188-f001:**
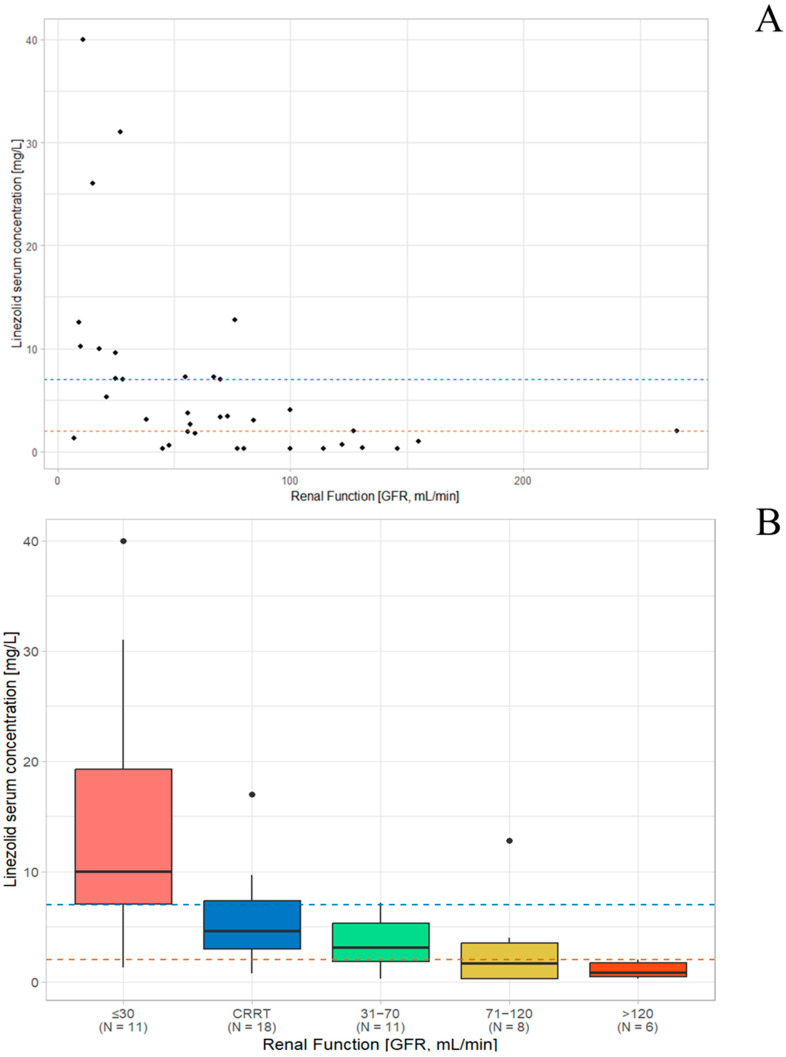
Correlation between linezolid serum concentrations (C_min-ss_) and renal function (**A**) Among non-CRRT patients; (**B**) Aggregated data by groups, including CRRT group. Blue line: Toxicity threshold for linezolid (7 mg/L); Orange line: Efficacy threshold for linezolid (2 mg/L). Abbreviations: CRRT = Continuous Renal Replacement Therapy, GFR: Glomerular Filtration Rate.

**Table 1 antibiotics-14-01188-t001:** Study population.

	N	Overall, N = 54	CRRT, N = 18	Non-CRRT, N = 36	*p*-Value
Sex	54				0.673
Female		16 (30%)	6 (33%)	10 (28%)	
Male		38 (70%)	12 (67%)	26 (72%)	
Weight (Kg)	54	77.00 (66.00, 90.00)	78.50 (70.0, 91.0)	76.00 (65.00, 89.00)	0.358
Height (cm)	54	172.5 (160.0, 178.0)	170.0 (160.0, 175.0)	175.0 (162.5, 179.00)	0.778
BMI (Kg/m^2^)	54	26.27 (24.22, 28.6)	27.77 (24.4, 30.8)	25.58 (24.15, 27.76)	0.180
Septic shock	54	27 (50%)	13 (72%)	14 (39%)	0.021
Organism	20				0.382
*E. faecium*		11 (55%)	3 (43%)	8 (61,6%)	
*E. faecalis*		4 (20%)	1 (14%)	3 (23%)	
*S. haemolyticus*		1 (5%)	0 (0%)	1 (7.7%)	
*S. aureus*		4 (20%)	3 (43%)	1 (7.7%)	
ALT (U/L)	41	26.00 (16.00, 49.00)	26.00 (13.00, 37.00)	36.50 (16.00, 68.00)	0.285
AST (U/L)	39	44.00 (26.00, 68.00)	39.00 (23.00, 66.00)	48.50 (28.00, 94.00)	0.507
Creatinine (mg/dL)	53	0.97 (0.60, 1.44)	0.89 (0.47, 1.19)	1.01 (0.72, 1.48)	0.273
Urea (mg/dL)	52	55.00 (31.00, 79.00)	45.00 (30.00, 68.00)	56.00 (32.00, 99.00)	0.226
Albumin (g/dL)	49	2.72 (2.31, 2.90)	2.77 (2.34, 2.85)	2.61 (2.29, 2.92)	0.682
Tot. Bilirubin (mg/dL)	48	1.11 (0.43, 2.90)	1.37 (1.04, 3.35)	0.74 (0.35, 1.99)	0.029
CrCl (mL/min)	36	58.00 (26.00, 92.00)	NA (NA, NA)	58.00 (26.00, 92.00)	
eGFR (mL/min)	26	81.50 (57.0, 121.00)	NA (NA, NA)	81.50 (57.0, 121.00)	
CI (L/min · m^2^)	37	3.50 (2.80, 4.10)	3.60 (2.67, 3.85)	3.45 (3.00, 4.60)	0.581
ICU stay (days)	54	21.50 (13.00, 37.00)	21.50 (16.00, 46.00)	21.00 (10.50, 36.50)	0.435

Values expressed as Median (Q1, Q3) or n (%). Abbreviations: ALT = alanine aminotransferase, AST = aspartate aminotransferase, BMI = body mass index, CI = cardiac index, CrCl = creatinine clearance, eGFR = estimated glomerular filtration rate, ICU = intensive care unit. Reference Ranges: ALT 3–45 U/L, AST 0–40 U/L, Creatinine: 0.51–1.17 mg/dL, Urea: 18–48 mg/dL, Albumin: 3.6–4.8 g/dL, Bilirubin (tot.): 0.25–1 mg/dL.

**Table 2 antibiotics-14-01188-t002:** CRRT vs. non-CRRT. (**A**) Comparison between CRRT and non-CRRT populations. (**B**) Comparison between CRRT patients and non-CRRT patients with GFR ≤ 30 mL/min.

**(A)**				
	**Overall, N = 54**	**CRRT, N = 18**	**Non-CRRT, N = 36**	***p*-Value**
C_min-ss_ (mg/L)	3.90 (1.30, 7.20)	4.60 (2.70, 7.60)	3.20 (0.85, 7.20)	0.279
Overdosing (N)	15 (27%)	4 (22%)	11 (31%)	0.519
In Range (N)	22 (40%)	11 (61%)	11 (31%)	0.031
Underdosing (N)	17 (31%)	3 (17%)	14 (39%)	0.097
**(B)**				
	**Overall, N = 29**	**CRRT, N = 18**	**GFR ≤ 30** **mL** **/min, N = 11**	***p*-Value**
C_min-ss_ (mg/L)	6.60 (4.10, 9.70)	4.60 (2.70, 7.60)	10.00 (7.00, 26.00)	0.007
Overdosing (N)	12 (41%)	4 (22%)	8 (73%)	0.018
In Range (N)	12 (41%)	11 (61%)	1 (9%)	0.008
Underdosing (N)	5 (17%)	3 (17%)	2 (18%)	>0.0999

Values expressed as Median (Q1, Q3) or n (%). Abbreviations: CRRT = Continuous Renal Replacement Therapy, GFR: Glomerular Filtration Rate.

**Table 3 antibiotics-14-01188-t003:** Extended infusion vs. continuous infusion. (**A**) Comparison between extended infusion (3 h) and continuous infusion. (**B**) Comparison between extended infusion (3 h) and continuous infusion among CRRT and non-CRRT populations.

**(A)**						
		**Extended Inf. (3 h),**		**Continuous Inf.,**		** *p* ** **-Value**
		**N = 35**		**N = 19**		
C_min-ss_ (mg/L)		2.70 (0.80, 5.30)		7.20 (4.00, 10.20)		0.002
Overdosing (N)		6 (17%)		9 (47%)		0.018
In Range (N)		16 (46%)		6 (32%)		0.313
Underdosing (N)		13 (37%)		4 (21%)		0.224
**(B)**						
	**CRRT**			**Non-CRRT**		
	**Extended Inf. (3 h),**	**Continuous Inf.,**	** *p* ** **-Value**	**Extended Inf. (3 h),**	**Continuous Inf.,**	** *p* ** **-Value**
	**N = 12**	**N = 6**		**N = 23**	**N = 13**	
C_min-ss_ (mg/L)	4.15 (2.45, 5.60)	7.10 (4.90, 7.70)	0.261	2.00 (0.30, 3.70)	7.20 (4.00, 12.50)	0.003
Overdosing (N)	2 (17%)	2 (33%)	0.569	4 (17%)	7 (54%)	0.056
In Range (N)	8 (67%)	3 (50%)	0.627	8 (35%)	3 (23%)	0.708
Underdosing (N)	2 (17%)	1 (17%)	>0.99	11 (48%)	3 (23%)	0.143

Values expressed as Median (Q1, Q3) or n (%). Abbreviations: CRRT = Continuous Renal Replacement Therapy.

## Data Availability

The datasets generated and/or analyzed during the current study are not publicly available, but are available from the corresponding author on reasonable request.

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
