# Peer review of "Linezolid Serum Concentration Variability Among Critically Ill Patients Based on Renal Function and Continuous Renal Replacement Therapy Administration"

_antibiotics, 2025, doi:10.3390/antibiotics14121188_

Round 1

Reviewer 1 Report

Comments and Suggestions for Authors

The authors have presented a well-designed study and analysis in critically ill patients requiring antibiotic therapy. The article is well written. A few minor suggestions are listed here:

1. Lines 88-89: The numbers and associated percentages are confusing here. Suggest to remove percentages which are based on the total number of patients in the hospital or ICU. Only percentages out of the total number of included patients (N = 54) are relevant here.

2. Table 1: Suggest to include in title what the values are (mean, median) and what the values in brackets are (IQR?). This will improve readability of Table 1.

3. Section 6 is named "Patents". This is unclear. Maybe the section can have a different title?

Comments on the Quality of English Language

English and typographical editing is required. There are multiple typos with sentences starting in small letters. Introduction section has some paragraphs with a single sentence.

Reviewer 2 Report

Comments and Suggestions for Authors

While the topic is of interest, it adds little value to the existing knowledge. Previous population model suggested for dose modification in patients with GFR <30 ml/min.

Specific comments

  1. L66-67 - Is it the 30% of the drug eliminated in urine ( meaning 30% of 30-40%) which is unchanged or 30% total drug eliminated.
  2. Table 1. Which criteria was used to define sepsis?
  3. Table 1. Use the term organism isolated instead of germ
  4. In the result section, all the information mentioned was mentioned in the tables as well. Remove redundancies.
  5. L113-116 It hardly makes any difference if there is a significant difference between Linezolid levels between CRRT and non-CRRT patients if the levels in any group is within normal range. It's not the difference but over or under levels which are clinically important in any subgroups.
  6. Table 2A, Table 3A - Remove column N, it has wrong value for each row.
  7. Figure 1A - This figure has only non-CCRT patients, while it should have included all.
  8. L122-124 - claims are too far fetched as sample size is small.
  9. Figure 1A - what does blue and orange lines and black dots represents? There is no foot note
  10. Figure 1B- there is no need of legend, colors are already showing the groups. also CRRT is not a category in renal function. It can't be clubbed here.
  11. Figure 3B - BMI and GFR information is redundant
  12. Either keep figure or table but not both. both carries same information.
  13. Discussion should be written as comparison of current results with the previous research but the authors just described the previous studies.
  14. L229 - The study claimed high risk of underdosing in ARC patients, while no where in the research was mentioned how many patients were having ARC. Also max. GFR reported in the table 1 was 121ml/min. This is not ARC. Similarly, L234 - the authors claimed risk is lower in patients with ESRD, where nowhere it was defined and mentioned how many such patients were there.
  15. There is no footnote below any table 

Reviewer 3 Report

Comments and Suggestions for Authors

Thank you for the opportunity to review the manuscript by Agliardi et al. This paper discusses the outcomes of a single-center retrospective study conducted in adults receiving intravenous linezolid treatment. Overall, two groups were evaluated: one on continuous renal replecement therapy (CRRT), and one with no renal support (stratified further by renal function). Linezolid concentrations were measured, and the attainment of pharmacokinetic targets in the two groups was compared. The manuscript contains 2 Figures, 3 Tables, and 20 references. The key finding of this research is that linezolid exposure is not worse in patients on CRRT than those in the comparator groups. Applying CRRT also resulted in more cases with linezolid trough concentrations classified by the authors as “overdosing”. The paper is well written, but important issues need to be addressed to make it clinically relevant.

Major concerns

  1. My first major concern is the consideration of 2-7 mg/L therapeutic range for evaluating the adequacy of the treatment. While I do understand the concept presented by the authors, I find this approach questionable since linezolid trough concentrations will correspond to different areas under the concentration-time curve when intermittent or when continuous infusions are applied (and, obviously, for all individuals in the extended-infusion group). The authors are requested to compare their findings to those described by Pea et al. (https://doi.org/10.1128/aac.00177-10), and to identify the limitations of their approach accordingly.
  2. My second major concern is related to Figure 2B and Tables 2 and 3, which I find confusing since non-CRRT patients comprise a group of patients with various renal functions, as demonstrated in Figure 1. As a result, I am not convinced by the information conveyed by Figure 2B and Tables 2 and 3. In my opinion, results obtained in CRRT patients should be compared to those obtained in patients with a creatinine clearance of <30 mL/min, and to those obtained in further groups, both for extended and continuous infusions. In addition, the authors claim that “the standard dose of 600 mg twice daily seemed adequate in most cases in terms of target attainment (2-7 mg/L), whereas this cannot be said for other patient groups, such as those with AKI or ARC” (lines 240-243). According to Figure 1B, target attainment cannot be expected for the GFR=71-120 mL/min group, either.
  3. Do the authors think linezolid dosing decisions should always be made after making an accurate evaluation of creatinine clearance? This seems quite important considering the remarkable differences between eGFR and creatinine clearance values (Table 1). Could the Cockroft-Gault formula or other formulas relying on serum creatinine measurements be useful?
  4. Based on the results, can an ROC analysis be performed to find a predictive value of either renal function, or the combination of renal function and the type of infusion for under- and overdosing?

Minor remarks

  1. Thank you for providing the details of CRRT, which I think is valuable to readers. I recommend moving the paragraph in lines 98-106 to the Materials and methods section.

  2. Please provide reference ranges for the laboratory tests (ALT, AST, creatinine, urea, albumin and bilirubin). AST elevation may be a result of in vitro haemolysis; do you have data that make it clear whether this occurred?

  3. Please specify whether patients not on CRRT received any renal support.
  4. Do the authors think patients with a creatinine clearance lower than 30 mL/min should receive renal support to attain therapeutic linezolid exposure, or should doses be decreased, or should intermittent linezolid infusion be applied? Could an algorithm be created in this sense for making clinical decisions?
  5. A dose of 600 mg linezolid BID was employed in this study. In my opinion, the dosing information should be clearly highlighted (perhaps even by including this in the title of the paper) since there is apparently no evidence that the findings can be extrapolated to other dosage regimens.
  6. The authors cite the systematic review by Liu et al (ref. 10), but do not mention the paper of Villa et al (doi: 10.1186/s13054-016-1551-7). Please include the evaluation of this work in the discussion.
  7. The abbreviations in Italian should be removed from the references.

Round 2

Reviewer 3 Report

Comments and Suggestions for Authors

Thank you for providing replies to the issues raised in my review report, and for making the respective modifications in the manuscript.

Some of the replies have not been included in the revised paper. I think these replies, just like the others provided, help readers understand the presented work. Therefore, I request the authors to amend the manuscript with the following thoughts they have shared:

  • We believe an accurate assessment of renal function would be crucial for optimized linezolid dosing. Because eGFR may overestimate renal  performance in acutely ill patients for several reasons, such as decreased creatinine load [doi:10.1093/ndt/gfh707], delayed creatinine accumulation [doi:10.1007/s00134-003-2078-3], and hemodilution [doi:10.1186/cc9004],
    dosing decisions are best supported by creatinine clearance measured directly—rather than by eGFR values alone. Formulas that estimate renal function using eGFR are often unreliable and should be applied with considerable caution for clinical decisions [doi:10.1186/cc12777, doi:10.1111/aas.14540]. In patients with impaired and variable renal function, it is reasonable to consider dose adjustment of linezolid based on TDM to ensure that plasma concentrations remain within the therapeutic range [doi: 10.1007/s00134-020-06050-1].
  • Unfortunately, the Summary of Product Characteristics (SmPC) and many current guidelines still state that dose adjustments are not required for linezolid, even in the presence of significant alterations in renal function. In our view, the most appropriate approach at this stage would be to initiate therapy with the standard dose and then promptly individualize it based on early therapeutic drug monitoring (TDM). Where TDM is not available,  referral to centers with clinical pharmacology services should be considered. Otherwise, we recommend close clinical and laboratory monitoring to detect any signs of linezolid toxicity (e.g., thrombocytopenia). We hope that more robust evidence will soon become available to support evidence-based decisions and enable more appropriate, individualized dosing for all patients from the outset. In our experience, both infusion modalities are adequate; however, continuous infusion may be preferable in patients with augmented renal clearance (ARC) and should be used with caution in cases of renal impairment—always within a TDM guided framework.

Also, thank you for providing the reference ranges of the laboratory assays. It looks like AST, urea and total bilirubin levels were often higher than the upper limit of the reference range, and all patient groups were hypoalbuminaemic. I recommend discussing the potential impact of these deviations, if any, on linezolid pharmacokinetics.
